# Electropolymerization of robust conjugated microporous polymer membranes for rapid solvent transport and narrow molecular sieving

Zongyao Zhou[1], Xiang Li[1], Dong Guo[1], Digambar B. Shinde[1], Dongwei Lu[1], Long Chen[2], Xiaowei Liu[1], Li Cao[1], Ammar M. Aboalsaud [1], Yunxia Hu[3] & Zhiping Lai [1✉]

Pore size uniformity is one of the most critical parameters in determining membrane separation performance. Recently, a novel type of conjugated microporous polymers (CMPs) has shown uniform pore size and high porosity. However, their brittle nature has prevented them from preparing robust membranes. Inspired by the skin-core architecture of spider silk that offers both high strength and high ductility, herein we report an electropolymerization process to prepare a CMP membrane from a rigid carbazole monomer, 2,2′,7,7′-tetra(carbazol-9-yl)-9,9′-spirobifluorene, inside a robust carbon nanotube scaffold. The obtained membranes showed superior mechanical strength and ductility, high surface area, and uniform pore size of approximately 1 nm. The superfast solvent transport and excellent molecular sieving well surpass the performance of most reported polymer membranes. Our method makes it possible to use rigid CMPs membranes in pressure-driven membrane processes, providing potential applications for this important category of polymer materials.

[1] Advanced Membranes and Porous Materials Center, Division of Physical Science and Engineering, King Abdullah University of Science and Technology (KAUST), Thuwal 23955-6900, Saudi Arabia. [2] Core Labs, King Abdullah University of Science and Technology (KAUST), Thuwal 23955-6900, Saudi Arabia. [3] State Key Laboratory of Separation Membranes and Membrane Processes, School of Materials Science and Engineering, Tiangong University, 300387 Tianjin, P. R. China. ✉email: Zhiping.lai@kaust.edu.sa

The membrane process is one of the most energy-efficient separation methods that can potentially save up to 90% of energy compared to conventional technologies such as distillation[1]. Membranes with a pore size in the range of 1–2 nm are classified as nanofiltration, which is an important membrane process that has found wide applications in the pretreatment of seawater desalination, organic solvent purification, and separations of fine chemicals[2–6]. Notably, the application of organic solvent nanofiltration (OSN) has received increasing attention in the chemical industry owing to the proliferating use of organic solvents, more strict environmental regulations and growing needs for energy savings[7,8].

Generally, membranes with a uniform pore size and high porosity offer high separation performance. Hence, membranes made of crystalline porous materials, such as zeolites, metal-organic frameworks (MOFs), and covalent organic frameworks (COFs), have achieved superior performance in gas and liquid separations[9–11]. However, these membranes are very difficult to prepare in large scales[12]. On the other hand, nanofiltration membranes made of conventional polymers, such as polyamide (PA), polysulfone (PS), and polymers of intrinsic microporosity (PIM), are easy to prepare and cheap but have a wide pore size distribution and low porosity which have comprised their performance[6]. Hence, to make polymer membranes that possess an equivalent porous structure as crystalline porous materials is currently one of the most promising directions in membrane research.

Recently, a novel type of conjugated microporous polymers (CMPs) has exhibited impressive pore size uniformity, high surface area, and high solvent resistance. The conjugated molecular structure further endows CMPs with interesting chemical, electrical and optical properties that have found potential applications as organic electronics, chemical sensors, and photovoltaic devices[13–20]. In principle, CMPs should also have good potential in membrane applications. We recently applied the CMP membrane in Li-S batteries as an efficient separator to block the shuttering effect of polysulfides but allow the fast transport of Lithium ions based on the molecular sieving effect[21]. However, unless the monomers are chemically modified with flexible segments to enhance the ductility[22], most of the reported CMPs are too brittle to form a robust membrane for pressure-driven membrane processes.

Spider silk is one of the strongest biological fibers that has both high strength and ductility. The excellent mechanical properties are due to its elegant nanocomposite structure, which is composed of a bundle of fibrils embedded in a non-crystalline matrix[23,24]. Inspired by the silk structure, here we report a novel strategy to enhance the mechanical properties of CMP membranes. A porous and conductive PDA-CNT support developed in our previous studies was used as a robust core scaffold[8], which was prepared from multiwall CNTs bound with polydopamine (PDA) by using a simple vacuum filtration method[25,26], and it features a smooth surface and an extremely high permeability. A polycarbazole (PC) CMP membrane was prepared inside the PDA-CNT network from a rigid conjugated monomer, 2,2′,7,7′-tetra(carbazol-9-yl)-9,9′-spirobifluorene (SpCz), using an electropolymerization (EP) process, yielding a robust CNT-EP-PC composite membrane. The SpCz monomer has a spiro center, which creates an intrinsic space around 1.15 nm inside the molecular structure, as illustrated by the DFT calculation (see Fig. S1 and Supplementary Information for details). The focal spiro center connects to four carbazole groups that serve as the polymerization linkages to form a 3D conjugated microporous polymer network. The SpCz consists of mainly aromatic rings. The resulting polymers are expected to have a uniform pore size in the nanofiltration range and be chemically stable and hydrophobic,

which are all desirable properties for OSN applications. In consequence, the synthesized CNT-EP-PC composite membrane showed excellent mechanical properties and "COF-like" high surface area and uniform pore size. The optimal membrane structure exhibited a high organic solvent flux for the application of OSN and a sharp molecular sieving in the separation of dye molecules, surpassing the performance of most reported polymer membranes thus far.

## Results

**Properties of the PDA-CNT support**. The structure of the PDA-CNT support is depicted in Fig. 1. The top-view SEM image (Fig. 1a) shows that the support has a similar buckypaper type of porous network. However, from the cross-section SEM image (Fig. 1b), some CNT bundles are vertically aligned, which is possibly due to the sucking effect of the vacuum filtration. The average layer thickness is around $300 \pm 22$ nm. The TEM image (Fig. 1c) reveals that the CNT has a multiwall structure. The diameter of the internal channel is around 2.2 nm, and the outside diameter is around 7 nm. The Raman spectra (Fig. 1d) exhibit three peaks, which are well consistent with the reported D band, G band, and 2D band of the CNT carbon structure. The XPS spectra (Fig. 1e) indicates that the surface contains oxygen, nitrogen, and carbon functional groups. Deconvolution of the C1s peak (Fig. 1f) unveils that the carbon signals come from both CNT and polydopamine, but mainly from the $sp2$ carbons of CNTs. As listed in Table 1, the PDA-CNT support has an average pore size of $21 \pm 3$ nm, electrical conductivity of $1.06 \pm 0.02 \times 10^3$ S/cm, and methanol permeance of $\sim 15,000 \pm 925$ L m$^{-2}$ h$^{-1}$ bar$^{-1}$ (LMH/bar). The permeance of the support is $\sim$2–3 orders of magnitude higher than that of regular polymer supports with similar pore size[27,28].

The PDA-CNT support was further characterized in Fig. S2 by atomic force microscopy, surface charge, thermogravimetric analysis (TGA), and nitrogen physisorption. The AFM image (Fig. S2a) shows a similar surface structure as the top-view SEM image and gives an average surface roughness of $13 \pm 1.6$ nm. A peak zeta potential of $-72$ mV was measured from the PDA-CNT aqueous suspension (Fig. S2b), suggesting that the CNT surface was highly negatively charged, which explains why a stable CNT suspension can be easily prepared and a uniform porous network can be obtained after vacuum filtration. The TGA analysis (Fig. S2c) shows a $\sim$20% weight loss from 350 to 500 °C due to the thermal decomposition of polydopamine. The nitrogen adsorption isotherm (Fig. S2d) showed a type IV curve according to the IUPAC classification. The Brunauer–Emmett–Teller (BET) specific surface area was quite low, only 53.7 m²/g. The pore size distribution calculated by the nonlinear density function theory (NLDFT) displayed a sharp peak at ~2.3 nm (Fig. S2e) and a broad peak ranging from 6 to 45 nm (Fig. S2f). The sharp peak matches the diameter of the CNT internal channel, while the broad peak should come from the network pores.

**The electropolymerization process to prepare CNT-EP-PC composite membranes**. The CNT-EP-PC composite membrane was prepared in a standard three-electrode electrochemical cell using the PDA-CNT support as a working electrode, as shown in Fig. 2a. The polycarbazole layer was grown by cyclic voltammetry (CV) scanning between $-0.8$ V and $1.03$ V (vs. Ag/Ag⁺). A scan rate of 0.05 V/s was used to ensure enough reaction time. In the first positive CV scan (Fig. 2b and Fig. S3a), an oxidative peak emerged at 0.77 V, which was related to the oxidation of carbazoles to radicals, as shown in the mechanism illustrated in Fig. 2c[20,21]. The carbazole radicals then coupled with each other to form dimeric carbazole cations during oxidation[18,19]. In the cathodic sweep scan, the dimeric carbazole cations were reduced

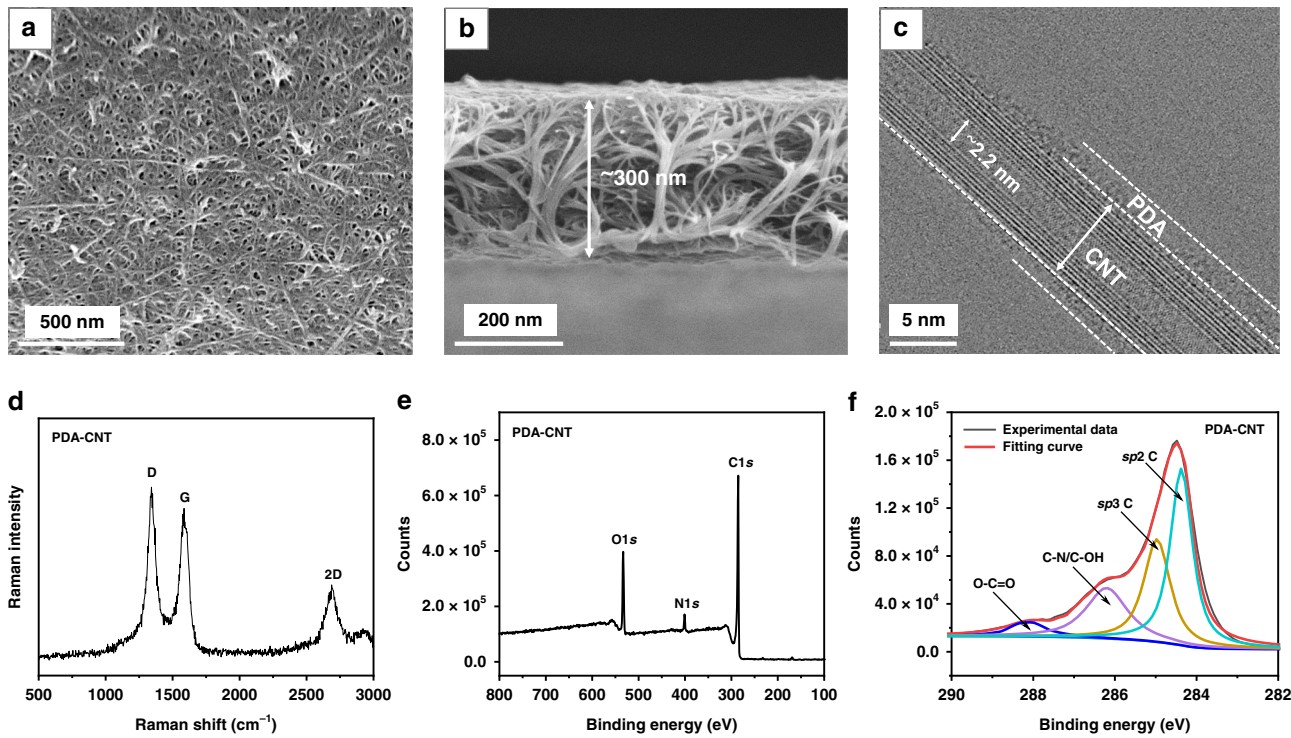

**Fig. 1 Characterization of the PDA-CNT support. a** Top-view SEM image. **b** Cross-section SEM image. **c** TEM image of a single CNT. **d** Raman spectra **e** XPS spectra of O, N, and C. **f** Deconvolution of the C1s XPS peak.

| Table 1 Properties of the PDA-CNT porous support. | | | |
|---|---|---|---|
| **Layer thickness (nm)** | **Average pore size (nm)** | **Electrical conductivity (S/cm)** | **Methanol permeance (LMH/bar)** |
| 300 ± 22 | 21 ± 3 | $(1.06 ± 0.02) × 10^3$ | $(1.5 ± 0.09) × 10^4$ |

to their neutral state, resulting in a reduction peak at 0.60 V. In the second and further scan cycles, both the oxidation and reduction peak currents increased with the number of cycles owing to the extension of polymerization and the membrane growth. Eventually, a highly crosslinked film was formed on the support[16]. Figure 2d shows the membrane thickness vs. the number of growth cycles. The thickness of the CNT-EP-PC composite membrane remained almost constant during the first five cycles. This is because the growth occurred inside the PDA-CNT porous support, which was confirmed by the SEM image that will be discussed in Fig. 3b. After that, the membrane thickness increased proportionally with the number of growth cycles. As a reference, the thickness of the polycarbazole layer grown on a dense indium tin oxide (ITO) support increased linearly from the beginning, and there was no incubation period of growth. It thus suggests again that the initial growth of the CNT-EP-PC membrane should have occurred inside the porous support network.

**Structures of CNT-EP-PC composite membranes**. The membrane prepared by 15 growth cycles, which is denoted as CNT-EP-PC15, was used as a benchmark system to characterize the membrane structures. The chemical structure of the as-synthesized CNT-EP-PC15 membrane was studied by Fourier transform infrared (FT-IR) spectroscopy (Fig. 2e). The three strong peaks observed in monomer SpCz at 720, 745, and 821 cm$^{-1}$ corresponded to the stretching frequencies of bi-substituted carbazoles.

In the CNT-EP-PC membrane, the peak at 720 cm$^{-1}$ shifted to 728 cm$^{-1}$; a new peak appeared at 801 cm$^{-1}$; and the peak intensities of 728 cm$^{-1}$ and 801 cm$^{-1}$ were almost the same. All these features were due to the polymerization of carbazoles to form a crosslinked framework, as reported in other polycarbazole polymers[20]. The dimerization process of carbazoles was further confirmed by $^{13}$C solid-state nuclear magnetic resonance (NMR) spectroscopy (Fig. 2f). The dimerization changed the carbazole rings from bi-substitution to tri-substitution, which caused the peak of the spiro carbon slightly shifted from 67.5 to 69.4 ppm, different peak intensities at ~111 and 142 ppm, and a relatively broad and less resolved spectra when compared the spectra of the CNT-EP-PC membrane with that of the monomer SpCz[29].

The top-view SEM image of CNT-EP-PC15 (Fig. 3a) shows a dense polycarbazole layer with a granular surface structure. The cross-section SEM image (Fig. 3b) shows that the polycarbazole layer was grown from both sides of the PDA-CNT support, forming a sandwich structure. The PDA-CNT support was not filled entirely by the polycarbazole layer. The central part remained almost empty. The overall thickness of the composite membrane was approximately 416 ± 19 nm. Taking account of the thickness of the PDA-CNT support, which is around 300 nm, it can be deduced that the structure of the CNT-EP-PC15 membrane contains three regions from the surface to the center: the top region with a thickness around 58 nm from both sides composes of a dense polycarbazole layer; the next intermediate region that is about 50 nm thick consists the polycarbazole and

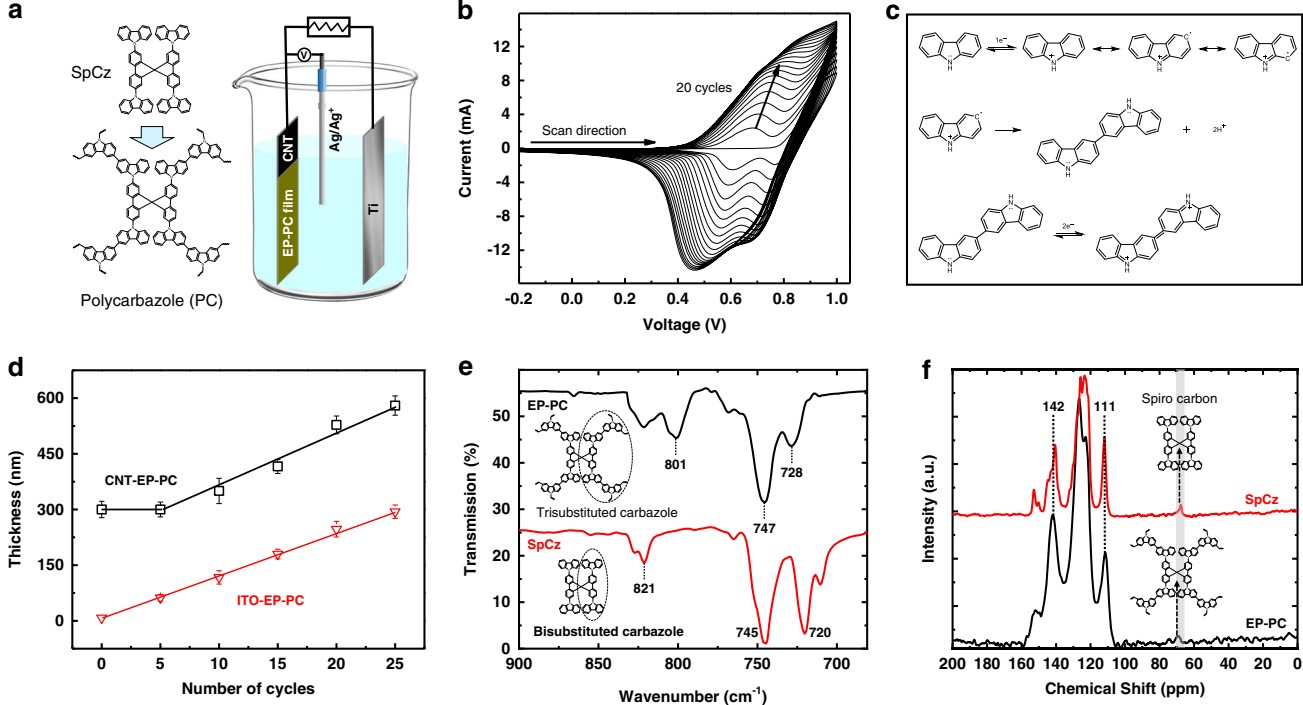

**Fig. 2 Membrane preparation and chemical structure. a** The setup of a three-electrode electrochemical cell for the electropolymerization process. **b** CV profiles of the electropolymerization process recorded for 20 scan cycles. **c** Mechanism of monomer oxidation, crosslinking, and reduction during the CV scans. **d** Membrane thickness vs. the number of CV scans. **e** FT-IR spectra of monomer SpCz and CNT-EP-PC15 membrane. **f** $^{13}C$ CP-MAS of monomer SpCz and CNT-EP-PC15 membrane.

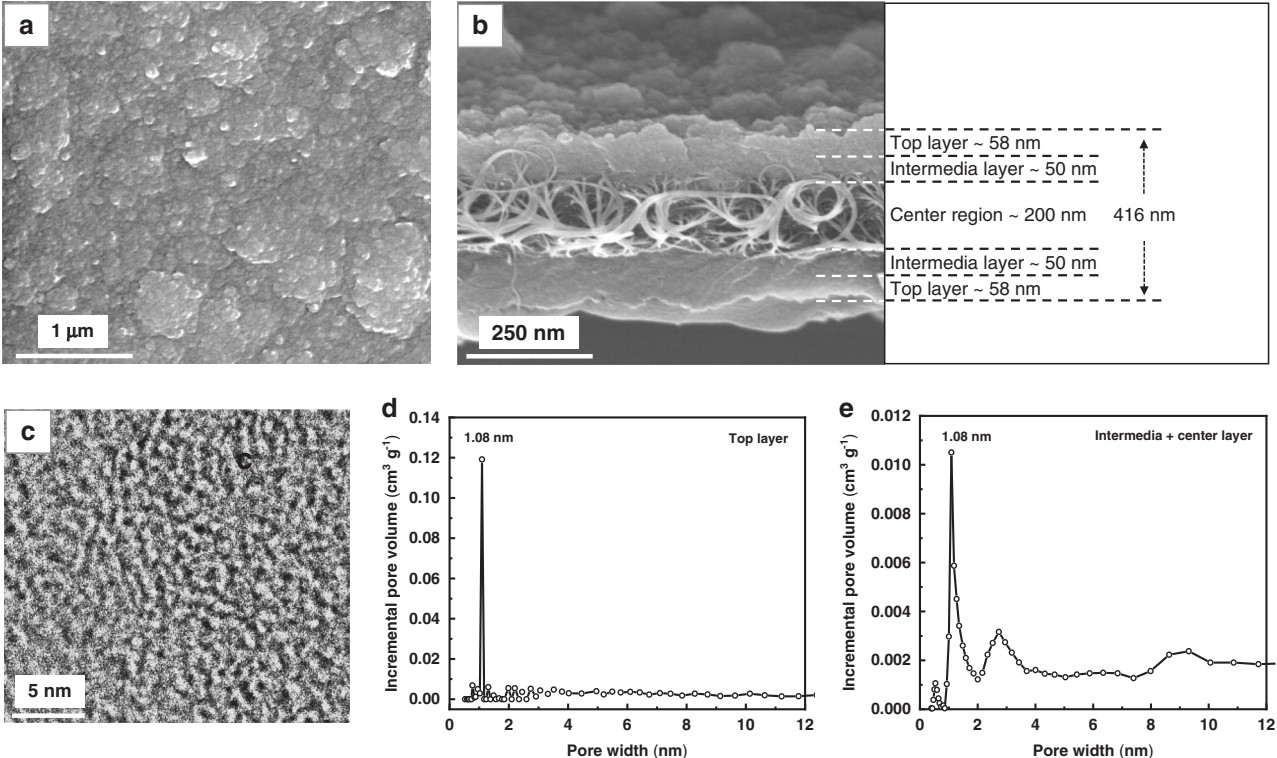

**Fig. 3 Membrane structure and pore size distribution of the CNT-EP-PC15 membrane. a** Top-view SEM image. **b** cross-section SEM image and the illustrated three membrane regions from top to the center. **c** TEM image of the top polycarbazole layer **d** Pore size distribution of the top polycarbazole layer. **e** Pore size distribution of the intermediate and central regions. The pore size distribution was calculated from the corresponding $N_2$ adsorption isotherm using the NLDFT method.

CNT composite structure; and the central region of about 200 nm remains the same structure as the PDA-CNT support.

The polycarbazole layer was found amorphous, as shown by the XRD pattern in Fig. S3d. The TEM image of the top polycarbazole layer (Fig. 3c) showed that it contained micropores with an average pore size of around 1 nm. The pores were randomly distributed, but the pore size was uniform. The TEM image in the intermediate region (Fig. S3c) showed that the polycarbazole layer was attached tightly first to the polydopamine layer and then to the CNT surface. The membrane pore size was further determined by nitrogen physisorption. The adsorption isotherms of the top region and the rest part of the membrane (i.e., intermediate region and central region) are shown in Fig. S3e and Fig. S3f, respectively. The BET specific surface area of the top region was as high as 548 m²/g, implying a highly microporous structure, but that of the rest part was significantly reduced to 255 m²/g, which was obviously due to the much low surface area of the PDA-CNT support. Figure 3d, e showed the pore size distributions of the top region and the rest part of the membrane that were calculated from the corresponding adsorption isotherms by the NLDFT method, respectively. The top polycarbazole layer exhibited a single sharp peak at 1.08 nm. The rest part also showed a sharp peak at the same pore size. Hence, we can conclude that this uniform pore size should come from the polycarbazole membrane. The rest part also contained pores ~3 nm, which was likely attributed from the CNT internal channels and large pores >8 nm that should come from the network pores of the support. The top polycarbazole layer should be the functional layer that determines the performance of the composite membrane. For this layer, both the surface area and the pore size distribution are equivalent to those of crystalline porous materials, such as COFs, zeolites, and MOFs[9,30].

**Mechanical properties and wettability of CNT-EP-PC composite membranes.** The mechanistic properties of the CNT-EP-PC15 membrane were studied first by the standard tensile test, and the stress vs. strain curve was shown in Fig. 4a. As shown in the insert image, a robust and flexible CNT-EP-PC15 membrane was formed on the PDA-CNT support, but as a comparison, the membrane grown on the dense indium tin oxide glass (ITO) support cracked easily and could not even be detached from the support, which was expected from the previous studies[22,31]. The CNT-EP-PC15 membrane showed a tensile strength of 26.5 MPa and a ductility of 18.9%. The Young's modulus is given by the slope in the linear region, which is ~2.2 GPa in Fig. 4a. These are excellent mechanical properties. The tensile strength of CNT-EP-PC15 is ~10–20 times stronger than that of normal polymer composite membranes as listed in Table S1[32–34]. As a result, the membrane can bear substantial bending and weight, as illustrated in Fig. 4b, c, supplementary movie 1, and supplementary movie 2. The Young's modulus of the membrane was further measured by the peak force quantitative nanomechanical mapping (PFQNM) method. The mapping curve is shown in Fig. 4d, and Young's modulus along the curve is shown in Fig. 4e. An average Young's modulus of ~3 GPa was obtained. Although the value is larger than that measured by the standard tensile test, they are in the same magnitude and can be considered consistent. The Young's modulus value is also ~5–8 times higher than that of the reported polyamide membranes and thus demonstrates the excellent mechanical properties of the CNT-EP-PC membrane again[35].

Figure 4f shows the wettability of the CNT-EP-PC15 membrane towards the water and organic solvents. The contact angle of water is ~150°, while that of methanol is almost 0°. Therefore, the membrane has a dual super-hydrophobicity and super-oleophilicity characteristic. This special wettability was further confirmed by dynamic contact measurements, as shown in Fig. S4 as well as in the supplementary movie 3, which showed that water droplets bumped off the membrane surface spontaneously like that on a lotus leaf, but methanol droplets spread immediately on the surface. The super-oleophilic surface is very useful because it can enhance the organic solvent flux[36,37]. This special wettability may be related to the conjugated aromatic polymer structure. The rough granular surface may further have enhanced this property[38].

**Organic solvent nanofiltration performance.** Three types of membranes, CNT-EP-PC10, CNT-EP-PC15, and CNT-EP-PC20 were tested, which were prepared from 10, 15, and 20 growth cycles and have thicknesses of 350 ± 17 nm, 416 ± 19 nm, and 528 ± 24 nm, respectively. The solvent permeances of the three membranes are plotted in Fig. 5a vs. the reciprocal of the solvent viscosity $(1/\eta)$. The permeance showed a linear relationship with $1/\eta$. As we explained in our previously reported crystalline 2D COF membranes[9], this is a distinct feature of a porous membrane because it indicates that transport is governed mainly by the pore-flow model. Compared to conventional dense polymer membranes[3], the highly porous CNT-EP-PC membrane offers a much faster transport pathway and thus a much higher membrane flux.

The molecular sieving capability of the CNT-EP-PC membrane was determined via the separation of different dyes that have different molecular weights and different charges: Congo red (CR, MW 697, negative charge), acid fuchsin (AF, MW 586, negative charge), rhodamine B (RB, MW 479, negative charge), crystal violet (CV, MW 408, positive charge), safranine O (SO, MW 351, positive charge), and natural red (NR, MW 289, neutral). Their molecular structures and estimated sizes are illustrated in Fig. S5. The concentrations of the dyes were measured by UV-Vis spectroscopy (Fig. S6). The rejection of dyes vs. their molecular weights was depicted in Fig. 5b, which showed a typical S-shaped rejection curve. The rejection rate was found mainly determined by the molecular weight but no clear relationship with the charge of dyes. The results for CNT-EP-PC15 and CNT-EP-PC20 were very close, but the data for CNT-EP-PC10 were consistently lower, indicating that the CNT-EP-PC15 membrane was not thick enough to eliminate defects. Thus, the CNT-EP-PC15 membrane offers the optimal performance among the three types of membranes. Based on the results of CNT-EP-PC15 and CNT-EP-PC20, the two important rejection parameters, the molecular weight retention onset (MWRO) at a 10% rejection rate and molecular weight cut-off (MWCO) at a 90% rejection rate, were determined to be ~300 and 540 Da, respectively. The difference between MWCO and MWRO is only 240 Da, which is equivalent to that of our reported crystalline COF membrane[9] but much smaller than other porous polymer membranes[35], confirming the excellent molecular sieving capability, which is expected from its uniform porous structure. The separation of mixed dyes is illustrated in a diffusion cell in Fig. 5c. Initially, the left chamber contained a 1:1 molar ratio of the CR and NR mixture, whereas the right chamber was filled with fresh methanol. The picture was taken after 1 day of diffusion. The right chamber turned red, which indicated that only NR diffused through the membrane, whereas CR was blocked. This was confirmed by the UV-vis analysis in Fig. 5d, in which the feed solution showed both CR and NR characteristic peaks, but the permeate solution showed only the NR peak.

Figure 5e shows the long-term test of the OSN performance of the CNT-EP-PC15 membrane for an CR/methanol mixed solution for up to 24 h. It took the system ~2 h to reach a steady state. Subsequently, stable permeance of ~28 LMH/bar and rejection rate of 95% was maintained over the entire study period,

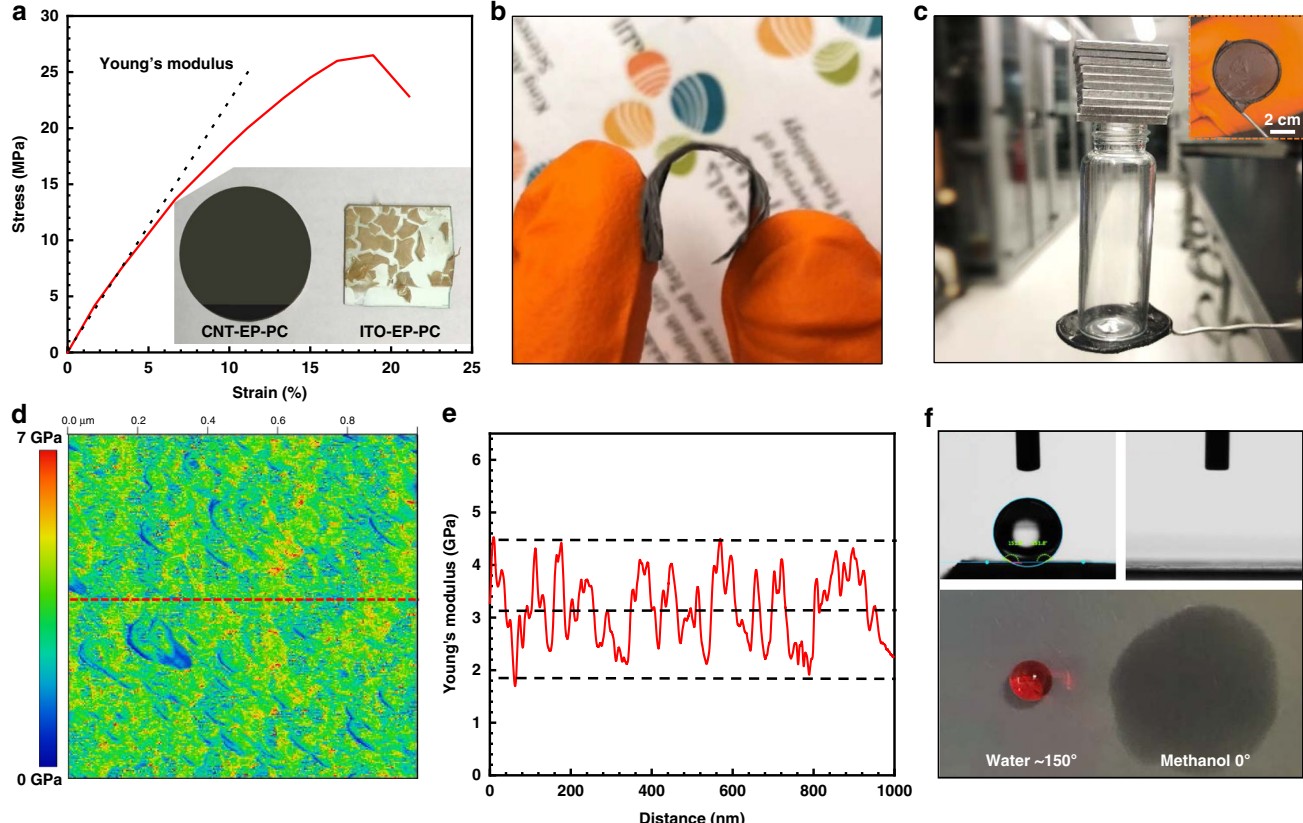

**Fig. 4 Mechanical properties and wettability of the CNT-EP-PC15 membrane. a** Stress vs. Strain curve measured by the standard tensile test. Inset: photographs of a CNT-EP-PC15 membrane and an ITO-EP-PC15 film fabricated on the dense ITO glass. **b** Photograph of the flexible CNT-EP-PC15 membrane; **c** Photograph of the CNT-EP-PC15 membrane holding a vial bottle and ten metal blocks. The total weight of the vial bottle and the metal blocks was about 8000-12000 times greater than the weight of CNT-EP-PC15 membrane itself. **d** AFM image showing the peak force quantitative nanomechanical mapping (PFQNM) curve. **e** The Young's modulus profile along the PFQNM curve; and **f** Contact angle measurements of water (left) and methanol (right) on the CNT-EP-PC15 membrane.

indicating that the membrane had not only high stability in the organic solvent but also antifouling potential. The long-term stabilities under other organic solvents were shown in Fig. S7. A similar trend was observed. It proved that the membrane was stable in acetonitrile, acetone, heptane, methanol, ethanol, and 2-propanol. Of note, all the reported permeances and rejection rates in this study were obtained during the steady-state period.

Figure 5f shows the comparison of the membrane performance with some of the best-reported membranes from literature[2,3,8,33,39–42]. The comparison is based on the methanol permeance. It can be seen that the performance of the CNT-EP-PC membrane is superior to most reported OSN membranes, including conventional polyamide thin film composite (PA-TFC) membranes[8,43,44], a 42-nm thick conjugated microporous polymer[35], a 2D $MoS_2$ membrane[45], and a MOF thin film nanocomposite membrane[39]. Only the 10-nm ultrathin polyamide membranes[3] and the ultrathin crystalline 2D COF membrane[9] showed better performance.

## Discussion
The polycarbazole membranes prepared from the rigid conjugated carbazole monomer, SpCz, exhibit a high surface area of 548 m$^2$/g and an almost uniform pore size around 1.08 nm. The surface area is lower than highly porous MOFs, but equivalent to other crystalline porous materials such as zeolite and COFs. The polycarbazole layer prepared on a dense ITO support is very brittle, which is consistent with literature reports and confirms that it cannot be used for pressure-driven membrane applications

in the stand-alone form. The PDA-CNT support has many advantages. Our thorough characterization results reveal that it is made of multiwall CNTs bound by polydopamine, generating a highly porous network structure with pore size around 21 nm. The CNTs form bundles that are vertically aligned inside the support due to the sucking effect of the vacuum filtration. Such a vertical alignment is expected to enhance the mechanical strength, improve the permeance, and facilitate a tight composite structure with the polymer. The permeance of the PDA-CNT support is 2–3 orders of magnitude higher than that of regular polymer supports with similar pore size, which offers the first benefit to the support since a highly permeable support will reduce the transport resistance and allow the monomers to diffuse easily into the network during the electropolymerization process to form a composite membrane. The second benefit is that the PDA-CNT has a smooth surface and a uniform structure. The reason is because the polydopamine modified CNTs have a large negative surface charge, which helps to form a stable and well-dispersed precursor suspension. The third benefit is that it is conductive, thermally stable, and mechanically strong due to the excellent electrochemical and mechanical properties of CNTs. All these benefits are key factors for the formation of robust, highly permeable, and compact composite membrane structures with polycarbazole during the electropolymerization process.

The CNT-EP-PC composite membranes are successfully prepared through the electropolymerization process. The membrane growth can be precisely controlled by the number of CV scans. Initially, the membrane is grown inside the PDA-CNT support.

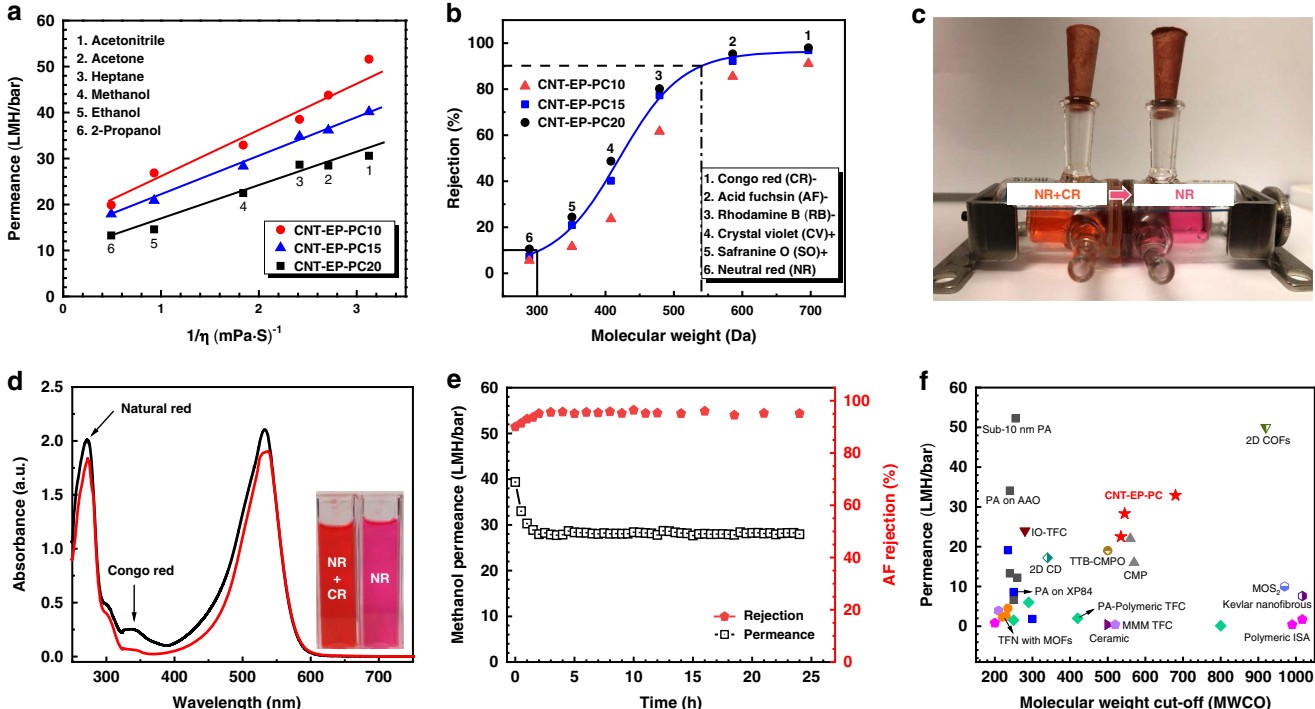

**Fig. 5 Performance of organic solvent nanofiltration and separation of dyes. a** Permeances of some common polar and nonpolar organic solvents under the transmembrane pressure drop of 1 bar vs. the inverse solvent viscosity on CNT-EP-PC membranes that have a thickness of 350 nm (CNT-EP-PC10), 416 nm (CNT-EP-PC15), and 528 nm (CNT-EP-PC20). **b** Rejection of dyes with different molecular weights under the transmembrane pressure drop of 1 bar. **c** Separation of mixed dyes by a CNT-EP-PC15 membrane through a diffusion cell. The chamber on the left side contained equal concentrations of Natural Red (NR) and Congo Red (CR) dyes, whereas the chamber on the right side was filled with pure methanol initially and turned red (NR) after 1 day of diffusion. **d** UV-vis spectra of the mixed dyes in the left chamber (black curve) and right chamber (red curve) of the diffusion cell in **c. e** Long-term filtration test of the CR/methanol solution on the CNT-EP-PC15 membrane. **f** Performance comparison of the CNT-EP-PC15 membrane with some best-reported nanofiltration membranes.

After ~5 CV cycles, the membrane starts to over grow, and the overall membrane thickness increases proportionally with the number of further CV cycles. The overall membrane structure comprises of three regions from the surface to the center: the top region composes of a dense polycarbazole layer; the intermediate region consists the polycarbazole and CNT composite structure; and the central region remains the same as the PDA-CNT support. The top polycarbazole layer should be the functional layer that determines the performance of the composite membrane.

The CNT-EP-PC composite membrane gives a tensile strength of 26.5 MPa, a ductility of 18.9%, and a Young's modulus approximately 2.2 GPa. The tensile strength is 10–20 times stronger than normal polymer membranes, and the Young's modulus is also 5–8 times higher than polyamide membranes. The excellent mechanical properties should allow the CNT-EP-PC membranes to be used in most pressure-driven membrane applications.

The organic solvent nanofiltration tests show that the permeance of the CNT-EP-PC membrane is inverse proportional to the solvent viscosity, which indicates that the solvent transport through the CNT-EP-PC membrane follows the pore-flow mechanism. A stable methanol permeance of 28 LMH/bar is achieved on the CNT-EP-PC15 membrane, which is superior to most reported OSN membranes. The rejection of dyes on the CNT-EP-PC15 membranes is found to be mainly related to the dye' molecular weight or molecular size, but no obvious relationship with their charges. The MWCO and MWRO of the CNT-EP-PC15 membrane are 540 and 300 Da, respectively. The narrow region between MWCO and MWRO indicates an excellent molecular sieving. The fast transport of organic solvents and

the narrow molecular sieving of dye molecules observed on the CNT-EP-PC membrane confirm that the excellent porous structure of CMPs, i.e., high surface area and uniform pore size, has good potential to achieve superior membrane separation performance. Furthermore, our long-term studies indicate that the CNT-EP-PC membrane is stable in most organic solvents. Our approach of using the electropolymerization process to form a robust composite membrane on the PDA-CNT support should be well extendable to most CMPs. Therefore, we expect that our approach will open a door for the potential application of this important category of polymer materials in pressure-driven membrane processes.

## Methods

**Materials**. CNT powder (5–30 μm length, purity >95%) was purchased from XFNANO, China. All other chemicals were purchased from Sigma–Aldrich and used as received.

**Preparation of the PDA-CNT support**. Followed our previously reported procedure[8], 100 mg CNT powder was sonicated in 1000 mL DI water for 1.5 h using a probe ultrasonicator (25 mm in probe diameter) under a power of 360 W. The mixture was centrifuged at 10,000 rpm for 40 min to remove any undispersed CNT powder. The supernatant was collected and heated to 40 °C. Then, 100 mg dopamine and 100 mL of 0.1 mol/L Tris buffer (pH = 8.5) were added into the supernatant under stirring. The mixture was kept stirring at 40 °C for 6 h. After that, the polydopamine modified CNT solution was centrifuged again for 40 min at 8000 rpm to remove any agglomerations. The collected supernatant was filtered under vacuum onto a polyethersulfone (PES) sacrificial substrate at a fixed amount per area of 0.5 ml/cm². The PES substrate has a nominal pore size of 0.22 μm. It was removed by dimethylformamide (DMAc) after filtration, yielding a free-standing buckypaper type of PDA-CNT porous network. The prepared PDA-CNT supports were dried thoroughly in a vacuum oven at 60 °C.

**Electrochemical polymerization process**. Forty milligram of SpCz and 1.55 g tetrabutylammonium hexa- fluorophosphate were dissolved in 40 mL anhydrous $CH_2Cl_2$ and $CH_3CN$ (4/1, v/v) mixture by simple shaking for 30 s to obtain a homogeneous electrolyte solution, which was then loaded into a standard three-electrode electrochemical cell that was attached to an electrochemical workstation (CH Instruments Inc., model 660 C). An $Ag/Ag^+$ nonaqueous electrode was used as a reference electrode, a dry free-stranding PDA-CNT support or ITO as a working electrode, and a 4 cm by 6 cm sized titanium metal plate as a counter electrode. Cyclic voltammetry was conduct in the range between −0.8 V and 1.03 V with a scanning rate of 0.05 V/s. The prepared membrane was washed with dichloromethane at least three times to remove unreacted monomers and then immersed in acetonitrile to remove electrolytes. Finally, the membrane was dried in a vacuum oven at 120 °C overnight.

**Characterization**. SEM images were taken from an FEI Magellan 400 microscope. High-resolution TEM images were acquired from an FEI Titan transmission electron microscopy. The membrane surface roughness was analyzed by AFM (Bruker Dimension Icon, Germany), which also had the peak force quantitative nano-mechanical mapping (PFQNM) function to probe the surface Young's modulus. The chemical structure was characterized by FT-IR (Thermo Fisher Scientific, Nicolet iS10, USA) and solid-state 13 C NMR (Bruker 500 MHz, Germany). Nitrogen physisorption was conducted at 77 K on a volumetric adsorption analyzer (Micromeritics ASAP 2420, USA). The test samples were evacuated at 393 K for 24 h prior to the measurements. A drop shape analyzer (Kruss, DSA100, Germany) was used to measure the contact angles through a sessile drop method. The drop volume is 2.0 µL. The dye concentration was measured by UV-Vis spectroscopy (Agilent, Cary 5000, USA). The mechanical strength of the membrane was measured by a universal mechanical testing machine (HY-0580, China) according to previous reports[46–48]. The zeta potential of the PDA-CNT suspension was measured by a Nanosize & Zeta Potential Analyzer (Litesizer 500, Anton Paar, Austria). The liquid extrusion porosimetry was measured by a capillary flow porometer (Porolux 1000, IB-FT GmbH Berlin, Germany). The electrical conductivity was measured by a Four-Point probe system (RTS-8, Probers Tech, China). The thermogravimetric analysis (TGA) was analyzed through a thermal gravity analyzer (TG 209 F3 Tarsus, Germany) in nitrogen at the heating rate of 10 °C/min. The Raman spectroscopy (XploRA PLUS, Horiba, Japan) of the PDA-CNT support was recorded in the wavelength range of 500–3000 cm$^{-1}$. The chemical state of the PDA-CNT surface was analyzed by X-ray photoelectron spectroscopy (XPS, K-alpha, Thermofisher, USA).

**Organic solvent nanofiltration (OSN) and separation of dyes**. The OSN tests were performed at room temperature using a commercial permeation cell (HP4750, Sterlitech®). The feed chamber has a suspending stir bar to keep the solution homogeneous. It was further connected to a solution tank to increase the total volume to be around 2.5 L to ensure the concentration change in the feed side during the permeation test is negligible. The pressure of the feed side was set to 1 bar, while the permeate side was open to the atmosphere. A series of dye solutions with a concentration of 100 ppm was used as feed solutions. The permeate was collected and the weight was monitored by a digital balance. The permeate collected in the first 2 h was decanted to allow the system to approach the steady state. The dye concentration was measured by a UV-vis spectrophotometer. The solvent permeance $P$ (LMH/bar) was calculated by Eq. (1),

$$P = V/(A \cdot \Delta t \cdot \Delta P) \tag{1}$$

where, $V$ (L) is the volume of the permeated solvent collected during a certain time period $\Delta t$ (h) under the pressure different $\Delta P$ (bar), and $A$ is the membrane area (m$^2$).

The dye rejection $R$ (%) was calculated by Eq. (2),

$$R = (1 - C_P/C_F) \times 100\% \tag{2}$$

where, $C_P$ and $C_F$ are the dye concentrations in the permeate and feed solutions, respectively.

## Data availability

The data that support the findings of this study are available from the corresponding author on reasonable request.

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

## Acknowledgements

The project was supported by King Abdullah University of Science and Technology under the competitive research grant URF/1/3769-01 and Baseline fund BAS/1/1375-01.

## Author contributions

Z.Z. designed the experiment and contributed to the majority of the experiment and data analysis. X.L., D.G., and Y.H. helped in membrane preparation. D.S. and D.L. contributed to the characterization and data analysis. L.C. conducted PFQNM measurement and data analysis. X.L., L.C., A.A., and Y.H. helped in data analysis and writing. The original draft was prepared by Z.Z., and reviewed, validated and edited by Z.L.

## Competing interests

The authors declare no competing interests.
