## [Peer Review File · Nature Communications]

REVIEWER COMMENTS

Reviewer #1 (Remarks to the Author):

The manuscript by Zhou et al. presents the design of a novel nanofiltration type membrane by specific electropolymerization across the surface of carbon nanotube based scaffolds. The work is of quality but several aspects shall be considered and revised prior to publication, particularly on the purpose statement, characterization of the pores within the materials and potentially also on the diffusion experiments.

1) The work is novel in its current approach but it is felt that the pathways and reactions for controlling porosity through such reactions is not new. This review highlights such previous strategies:

<https://link.springer.com/article/10.1007/s10008-015-2973-x>

The rationale for the choice of monomers to yield specific morphologies or pore size distributions is equally not clear. It is suggested that the authors redevelop that rationale to better justify their choices in the introduction by backing themselves better on previous works.

2) There are contradicting statements in the manuscript. An example lies on lines 82-85 on the thickness of the materials where it is first stated that it is constant for the first 5 cycles prior to being stated that it is not... this is unclear and proper ellipsometry data should be shown and discussed here. Other statements are approximate and unclear across the manuscript requiring revision.

3) The nature and surface state of the CNT scaffold is not well described or presented. It is understood that the authors present a bucky-paper type structure, corresponding to a non-woven type structure. These materials have been developed 10+ years ago and used as membranes for a number of applications (desalination, ultrafiltration type applications etc) and it would be extremely relevant to present previous works in this space since the purpose of the work here appears to be solely focused on their surface modification. A review here could help in the discussion and to identify key papers.

<https://www.mdpi.com/1996-1944/3/1/127>

Various properties of the CNTs (besides the TEM/AFM provided) should be given and both Raman/XPS would be required to characterize properly the scaffolds. Properties such as the pore size or porosity of the CNT scaffold and PDA-coated CNTs shall also be provided. The gain in pore size arising from the support would be substantial and it is important to show the changes relatively to the scaffolds. In addition, CNT buckypapers are typically rough since the roughness of the material is on the order of several CNTs in thickness due to the entanglement of the CNTs in the structure (as visible from the AFM data in sup mat and Figure 2). For this reason characterizing clearly the electric conductivity and other bulk properties and surface states (oxidation levels etc) of the scaffolding materials would be important to understand the polymerization pathways. I would not expect the ITO to behave like the CNTs...

3) The key challenge in the paper is however the characterization of the porosity and pore size distribution. It is unlikely that BJH tests can yield such sharp pore size distributions due to the

contribution of the CNTs in the overall adsorption process. More data on reference samples providing the bare CNTs and potentially different series of electropolymerization cycles shall be provided. It is visible that the pore size of the CNT scaffolds lies within a 100 nm or so. This would affect the physisorption data used for the calculation of the entire sample (CNT + polymer). Another reference sample should be done on the polymer alone (potentially on ITO). However a key challenge here is the mass determination! One would need a large volume of sample to perform this test properly and since no experimental details are provided on how it was done, further investigations shall be done to provide controls. This point is critical since the claim of the paper falls against this unique test.

3) In order to better characterize the porosity of the material it is suggested to not only perform physisorption but also chemisorption with various solvent vapours (or gases prone to chemisorption). It would also be extremely relevant to show TEM data (since the authors have access to it to confirm the pore size) or else look at non invasive techniques such as SAXS/WAXS or SANS. If indeed the pore size distribution is this sharp, the scattering patterns should be very clear. There are plenty of literature on such nanocomposites (CNT/MOFs or COFs type materials) which they authors can use here.

4) The mechanical testing are also unclear. The data shown is not clearly showing the thickness of the sample used. As per the BET data this raises concerns that the claims done for the samples tested are not appropriate. The authors should carefully check this and ensure that the image (2c) and mechanical testing data shown do correspond to the sample shown in the SEM etc. Also it would be interesting to show the surface of the samples after mechanical testing to assess if the COF-like structure developed would sustain the stress and not shatter. This would provide try indication on the mechanical stability of the material.

5) The permeation /diffusion tests are also confusing. The picture showed in Figure 3d shows a diffusion cell. Such type of cell cannot be pressurized to a large extent. It is therefore guessed that the cell was only used for dye diffusion and not the other tests of solvents permeation. This aspect is however not clear in the manuscript and shall be revised. If this is indeed the case the caption should be corrected and water diffusion data also provided. Looking only at the dye concentration across the material would not be sufficient in this case to show the true transport number for each molecule. Given the pore size osmotic effects could start to arise and this aspect is not discussed at all. Again previous works on CNT nanocomposite membrane materials used for osmosis should be considered, reviewed and discussed. One of such paper is provided for information and discussion: <https://www.sciencedirect.com/science/article/pii/S0376738812006977>

6) The diffusion experiments of dyes are particularly interesting and indeed provide some insights on the potential of this material. It would be relevant to however further discuss the molecular weight cut off (MWCO) of the materials. The MWCO is typically defined as the MW at which >80% rejection is achieved. here the MWCO falls on rhodamine B which has a MW of approx. 479 Da. This is rather small but likely larger than the 1 nm claimed by the authors. This aspect would likely be related to the fact that the pore size distribution is not as narrow as evaluated by BJH and that hydration (the material is hydrophilic as it seems) would likely swell/affect the pore size distribution. The authors should also characterize the streaming potential of the material to evaluate its zeta potential and

IEP. it is unclear at which pH these diffusion tests were performed and this would have a massive impact on rejection!

The wettability of the membranes for the different solvents should also be reported as well as the uptake (chemisorption) for each. from gravimetric methods. The CNTs are unlikely to be affected but the COF-like layer could. SEMs after filtration would also be required to assess the stability while long term filtration stabilities for the series and not only methanol should be provided.

7) On Figure 3 b please indicate clearly in the caption the thickness used for the calculation of the permeability. As a comment naming samples by code names makes the reading of the manuscript difficult and one should relate back to the synthesis conditions more often for clarity. Figure 3f could be updated with other relevant papers in this space and the benchmarking appears to be partial.

8) Overall the experimental details provided are insufficient and complete methods should be provided in supplementary materials.

Reviewer #2 (Remarks to the Author):

This manuscript reports a novel fabrication methodology, which enables membranes that exhibit impressive mechanical stability, promising organic solvent nano filtration performance, and molecular sieving capabilities. All the claims made are supported by convincing data and analysis. Further characterization of the PC-CNT interfaces would be a welcome (but not necessary) addition to this work as it obviously is the key for the high performance achieved. Considering that this is a highly competitive field of current and emerging significance, timely publication is essential. I recommend acceptance in its current form. I anticipate that this work will generate topical and broad interest as well as stimulate innovations and follow up studies.

Reviewer #3 (Remarks to the Author):

The authors prepared a rigid Polycarbazole (PC) membrane on the polydopamin modified CNT support. Indeed, the high physical property including mechanical and chemical stability is an important issue in polymeric membrane because of the pressure-driven process. The membrane performance is quite impressive, although it is not the best. Recently, the authors reported the similar membrane in Nano Energy, and the membrane was applied in Li-S battery. The authors incorporated polydopamin in the previous CNT frame to PC membrane, and applied to solvent separation membrane. Overall, this manuscript has to be substantially modified to be considered in nature communication, as below.

1. Similar concept was recently published to Nano Energy by same authors, although it was employed in Li-S battery membrane. First of all, the recent paper should be cited in this manuscript, and should to be described the different between the system and component of the previous and present papers. "Electropolymerization growth of an ultrathin, compact, conductive and microporous (UCCM) polycarbazole membrane for high energy Li-S batteries" (Nano energy 2020)
2. Very importantly, the authors only showed the physical properties of the polydopamin modified

CNT support. Because the physical properties of the composite membrane is more important, it is essential to show the physical properties of CNT-EP-PC composite membrane.

3. Also, the authors should show the membrane performance in the pressure applied system.

4. The PC film on the CNT frame exhibits narrow pore distributions. How about other PC-based membranes that have been reported previously? This has to be compared in the text and Figure 4f.

5. Please explain why the CNT-polydopamin support shows the oriented dendritic structure. Are the structure and orientation of the CNT support influenced by the thickness and pore size distribution of the PC layer?

Point to Point Response to Reviewers' Comments

Reviewer #1 (Remarks to the Author):

The manuscript by Zhou et al. presents the design of a novel nanofiltration type membrane by specific electropolymerization across the surface of carbon nanotube based scaffolds. The work is of quality but several aspects shall be considered and revised prior to publication, particularly on the purpose statement, characterization of the pores within the materials and potentially also on the diffusion experiments.

Response: We appreciate the reviewer's precious time to provide this constructive review, which is of great help to improve the quality of this manuscript. We have made a major revision and provided additional results to address the reviewer's concern. Detail explanations are provided in the below point-to-point responses.

Question 1: *The work is novel in its current approach but it is felt that the pathways and reactions for controlling porosity through such reactions is not new. This review highlights such previous strategies:<https://link.springer.com/article/10.1007/s10008-015-2973-x>. The rationale for the choice of monomers to yield specific morphologies or pore size distributions is equally not clear. It is suggested that the authors redevelop that rationale to better justify their choices in the introduction by backing themselves better on previous works.*

Response: We thank the reviewer for the suggestion of the review article. It enhanced our understanding of the mechanism of the electrochemical polymerization as a new approach to making high-performance membranes. We specified the rationale for the choice of monomer in the manuscript in Line 68-76.

Question 2: *There are contradicting statements in the manuscript. An example lies on lines 82-85 on the thickness of the materials where it is first stated that it is constant for the first 5 cycles prior to being stated that it is not... this is unclear and proper ellipsometry data should be shown and discussed here. Other statements are approximate and unclear across the manuscript requiring revision.*

Response: The contradictive statements are due to our poor writing. The growth curve of the polycarbazole membrane on the PDA-CNT support is different from that on the dense ITO support in the initial 5 cycles. The reason is that the growth on the PDA-CNT support occurred inside the porous PDA-CNT support during the initial stage, and hence the overall membrane thickness did not change during this stage. However, the ITO support is dense, so the total membrane thickness increases from the beginning of the growth. We confirmed the initial growth inside the PDA-CNT support by SEM observation. We revised the description in this part in Line 127-134.

For the suggested ellipsometry method, since we used the SEM to measure the membrane thickness directly, it is more accurate than ellipsometry. To minimize the experimental error, more batches of CNT-EP-PC membranes were prepared for the thickness measurement, and the results were updated in Fig. 2d.

Question 3: *The nature and surface state of the CNT scaffold is not well described or presented. It is understood that the authors present a bucky-paper type structure, corresponding to a non-woven type structure. These materials have been developed 10+ years ago and used as membranes for a number of applications (desalination, ultrafiltration type applications etc) and it would be extremely relevant to present previous works in this space since the purpose of the work here appears to be solely focused on their surface modification. A review here could help in the discussion and to identify key papers. <https://www.mdpi.com/1996-1944/3/1/127>*

Various properties of the CNTs (besides the TEM/AFM provided) should be given and both Raman/XPS would be required to characterize properly the scaffolds. Properties such as the pore size or porosity of the CNT scaffold and PDA-coated CNTs shall also be provided. The pore size arising from the support would be substantial and it is important to show the changes relatively to the scaffolds.

In addition, CNT bucky papers are typically rough since the roughness of the material is on the order of several CNTs in thickness due to the entanglement of the CNTs in the structure (as visible from the AFM data in sup mat and Figure 2). For this reason characterizing clearly the electric conductivity and other bulk properties and surface states (oxidation levels etc) of the scaffolding materials would be important to understand the polymerization pathways. I would not expect the ITO to behave like the CNTs.

Response: We agree with the reviewer's comments. The property of the PDA-CNT is indeed important to the success of the CNT-EP-PC composite membrane. Our current understanding is that it must be highly permeable to allow the polymer membrane to grow from inside the support to form a tight composite structure. Therefore, although the PDA-CNT support seems to have the buckypaper type of structure, we need to home make it following the procedure described in the method part, but not to use the commercial ones.

Following the reviewer's suggestion, we thoroughly characterized the PDA-CNT support in terms of the surface roughness by AFM (Fig. S2a), Raman spectra (Fig. 1d), chemical state by XPS (Fig. 1e and Fig. 1f), the pore size by liquid extrusion porosimetry (Table 1), thermal gravimetric analysis (Fig. S2c), and gas adsorption (Fig. S2d, Fig. S2e, and Fig. S2f). We included these results in Fig. 1, Fig. S2, and Table 1. We also discussed their possible effects on the membrane properties in Line 81-116.

Question 4: *The key challenge in the paper is however the characterization of the porosity and pore size distribution. it is unlikely that BJH tests can yield such sharp pore size distributions due to the contribution of the CNTs in the overall adsorption process. More data on reference samples providing the bare CNTs and potentially different series of electropolymerization cycles shall be provided.*

It is visible that the pore size of the CNT scaffolds lies within a 100 nm or so. This would affect the physisorption data used for the calculation of the entire sample (CNT + polymer). Another reference sample should be done on the polymer alone (potentially on ITO). However a key challenge here is the mass determination! One would need a large volume of sample to perform this test properly and since no experimental details are provide on how it was done, further investigations shall be done to provide controls. This point is critical since the claim of the paper falls against this unique test.

Response: Indeed, the pore size and porosity are not easy to determine in the current study. The reason is that the membrane is amorphous and the pore size is so small, which prevents the use of many structure determination methods such as wide-angle XRD or small-angle XRD. We studied the membrane by TEM. The result is included in Fig. 3c. Although in the TEM image we can identify the pores, the resolution is limited. In our humble opinion, the gas physisorption is probably the only reliable method that can determine the pore size in this pore size range. As mentioned by the reviewer, one challenge is to collect enough samples. We worked very hard and made hundreds of samples, as shown in the figure below, to overcome this issue.

Figure A: Pictures of a number of CNT-EP-PC15 membranes prepared for the gas adsorption studies.

Unless explicitly specified, all the characterization and performance tests were conducted on the CNT-EP-PC15 membrane in which the number 15 denotes the number of growth cycles. The reason is that this type of membranes gives relatively optimal results than samples with other thicknesses. From the structure analysis, it shows that this membrane contains three layers: the top layer contains a pure polymer film, the intermediate layer where the polymer forms a tight composite structure with the CNT, and the central region in which there is no polymer. The top layer is essentially the function layer that controls the permeation performance of the entire composite membrane. Therefore, we tried to detach the top layer from the rest part of the membrane shown in Figure A and studied the pore size and pore size distribution separately using

nitrogen physisorption (Fig. 3d and Fig. 3e in the revised manuscript). In this way, we successfully eliminate the interference of the PDA-CNT support. The pore size of the top layer is the same as the pore size of the PC membrane in the intermediate layer. We also compared the porous structure of the top layer with the PC membrane grown on the ITO support. The results are also well consistent with each other.

Question 5: *In order to better characterize the porosity of the material it is suggested to not only perform physisorption but also chemisorption with various solvent vapours (or gases prone to chemisorption). It would also be extremely relevant to show TEM data (since the authors have access to it to confirm the pore size) or else look at non invasive techniques such as SAXS/WAXS or SANS If indeed the pore size distribution is this sharp, the scattering patterns should be very clear. There are plenty of literature on such nanocomposites (CNT/MOFs or COFs type materials) which they authors can use here.*

Response: We thank the reviewer for the suggestions. As we explained in the previous response, although we can identify the pores by TEM, the resolution is limited because the membrane is amorphous. It also prevents the use of other diffraction-based techniques such as SAXS/WAXS or SANS. This is very different from crystalline materials such as CNT/MOFs or COFs.

To the best of our knowledge, we also believe the chemisorption method suggested by the reviewer will not allow us to have a better estimation of the pore size. First, the chemical adsorption is also an indirect method. Second, all these methods are essentially based on the Kelvin equation to determine the pore size. It needs a detailed model (such as Langmuir, BJH, NLDFT) to count for the surface interaction and surface inhomogeneity. The N₂ adsorption isotherm and the NLDFT model are currently the most widely used technique to determine the pore size from microporous to mesoporous range.¹⁻³ Unless the pore size is less than the molecular diameter of N₂, it is typically not recommended to use other species since it will require the development of a whole set of the model parameters from scratch, which is not possible particularly for an unknown system. Thus, in our humble opinion, the gas adsorption is the only reliable method in the current stage.

Question 6: *The mechanical testing are also unclear. The data shown is not clearly showing the thickness of the sample used. As per the BET data this raises concerns that the claims done for the samples tested are not appropriate. The authors should carefully check this and ensure that the image (3c) and mechanical testing data shown do correspond to the sample shown in the SEM etc. Also it would be interesting to show the surface of the samples after mechanical testing to assess if the COF-like structure developed would sustain the stress and not shatter. This would provide try indication on the mechanical stability of the material.*

Response: In the revision, we have made it clear that all the characterizations are conducted on CNT-EP-PC15 type of membranes unless explicitly specified. The CNT-EP-PC15 membrane is

prepared by 15 cycles of growth and has a thickness of around 416 ± 19 nm. Thus, we confirmed that membranes used in Fig. 3c (it is now Fig. 4c in the revised version), and other characterization tests, are the same type of membranes and the results are consistent with each other.

The mechanical test was carried out by two methods: the standard tensile test which gives the stress vs. strain curve in Fig. 4a, and an advanced method called peak force quantitative nanomechanical mapping (PFQNM) which gives the Young's modulus profile of the membrane surface in Fig. 4e. These two sets of data can compare with each other for data verification. As shown in our results, the Young's modulus obtained from the stress vs. strain curve is about 2.2 GPa, while the PFQNM method gives an average value of 3 GPa. Although there is about 40% deviation between the two methods, the values are in the same magnitude and can be considered consistent with each other. The tensile test is a destructive method, so the sample is totally damaged after the test. However, in other mechanical property tests, the membrane can be fully recovered when the applied stress is released. For example, Fig. B shows the SEM image of the membrane surface after the PFQNM test. No damage was found on the surface.

Figure B: Surface SEM image of CNT-EP-PC15 after PFQNM measurement.

Question 7: *The permeation /diffusion tests are also confusing. The picture showed in Figure 4d shows a diffusion cell. Such type of cell cannot be pressurized to a large extent. It is therefore guessed that the cell was only used for dye diffusion and not the other tests of solvents permeation. This aspect is however not clear in the manuscript and shall be revised. If this is indeed the case the caption should be corrected and water diffusion data also provided. Looking only at the dye concentration across the material would not be sufficient in this case to show the true transport number for each molecule. Given the pore size osmotic effects could start to arise and this aspect is not discussed at all. Again previous works on CNT nanocomposite membrane materials used for osmosis should be considered, reviewed and discussed. One of such paper is provided for information and discussion:*

<https://www.sciencedirect.com/science/article/pii/S0376738812006977>

Response: Yes, the diffusion cell shown in Fig. 4d (it is now Fig. 5c in the revised manuscript) was used only for a visual demonstration of the separation of mixed dyes as it is transparent. As described in the method part, all the OSN tests and rejection of dyes were conducted on a commercial metal permeation cell (HP4750, Sterlitech®) under applied pressures. We specified the applied pressure in the figure captions.

The consideration of the osmotic pressure is a good point. We estimated the effect as follows. The weight concentration of the dye solution in our studies is ~ 100 ppm. The molecular weights of the dyes are in the range of 300 ~700 Da. So the molar concentration is $< 3 \times 10^{-4}$. Assuming the dye molecules do not dissociate in organic solvents, the Van't Hoff factor is 1. So the osmotic pressure of the dye solution is,

$$\pi = iCRT < 0.006 \text{ bar}$$

During the experiments, the change in the feed concentration is $< 10\%$, and thus the change in the osmotic pressure due to the concentration rise is $< 6 \times 10^{-4}$ bar. Compared to the applied pressure (1 bar), the effect of the osmotic pressure is negligible.

The reviewer's suggestion will be very valuable in the study of reverse osmosis of organic solvents or commonly named as hyperfiltration. In this application, the osmotic pressure can be over 100 bar. We expect our EP membrane will be able to apply to this application in the future.

Question 8: *The diffusion experiments of dyes are particularly interesting and indeed provide some insights on the potential of this material. It would be relevant to however further discuss the molecular weight cut off (MWCO) of the materials. The MWCO is typically defined as the MW at which $>80\%$ rejection is achieved. here the MWCO falls on rhodamine B which has a MW of approx. 479 Da. This is rather small but likely larger than the 1 nm claimed by the authors. This aspect would likely be related to the fact that the pore size distribution is not as narrow as evaluated by BJH and that hydration (the material is hydrophilic as it seems) would likely swell/affect the pore size distribution.*

The authors should also characterize the streaming potential of the material to evaluate its zeta potential and IEP. it is unclear at which pH these diffusion tests were performed and this would have a massive impact on rejection!

The wettability of the membranes for the different solvents should also be reported as well as the uptake (chemisorption) for each. \from gravimetric methods. The CNTs are unlikely to be affected but the COF-like layer could. SEMs after filtration would also be required to assess the stability while long term filtration stabilities for the series and not only methanol should be provided.

Response: We have measured the MWCO in Fig. 5b. The value is 540 Da. The estimated molecular size of Rhodamine B is larger than 1 nm, as shown in Fig. S5. The reason why it can

pass through the membrane that has a pore size of 1.08 nm can be explained as follows. First, the kinetic diameter of complex molecules like Rhodamine B may be significantly smaller than their geometric size. Second, the structure flexibility of the polymer membrane may allow larger molecules to enter the pores. This is very common even in framework materials. For example, the pore size of a metal-organic framework material, ZIF-8, has a pore size of 0.34 nm, but it allows the transport of propylene, which has a kinetic diameter of 0.42 nm.⁴ The third reason is swelling, as mentioned by the reviewer. In real cases, all these factors may play a role. In this study, we used a practical approach to probe the effective pore size. As shown in Fig. 5b, we used a number of dyes that have different molecular weights and different charges to obtain the rejection vs. the molecular weight curve. The rejection curve showed a narrow transition range between 300 Da and 540 Da, which indicates that the membrane can achieve sharp molecular sieving.

Figure 5b: Rejection of dyes with different molecular weights and charges.

As shown in Fig. 5b, the studied dyes have different charges. The Congo Red, Acid Fuchsin and Rhodamine B are anionic dyes, Crystal Violet and Safranin O are cationic dyes, and Neutral Red is a neutral dye. We also measured the surface zeta potential at different pH and the results are shown in Fig. C. From the zeta potential diagram, the IEP was determined around pH 5.5. However, the zeta potential was measured in aqueous solution, while the dye separation in OSN was done in organic solvents. The rejection curve in Fig. 5b is primarily determined by the molecular weight; no relationship with the surface charge was identified. This is possible because the ionic dyes do not dissociate in organic solvents, and the surface charge measured in Fig. C doesn't apply in organic solvents as well. However, we agree with the reviewer that surface charge is a very important factor in membrane separation and we will conduct a systematic study on this topic in aqueous solutions in the future.

Figure C: Surface zeta potential of CNT-EP-PC15 membrane in aqueous solution

As suggested by the reviewer, we measured the contact angle for different organic solvents and the results are shown in Fig. D. The membrane showed the super-hydrophobic and superoleophilic wettability. The water contact angle is greater than 150° , but the contact angles to all the studied organic solvents are close to 0° . This wettability is beneficial for organic solvent nanofiltration.

We also tested the long-term stability of the membrane in different organic solvents. The results are included in Fig. S7. Good stability was demonstrated in all the studied organic solvents. The membrane after the long-term test was inspected by SEM, as shown in Fig. E. Compared to the SEM image before the test, no difference was noticed.

Figure D: Pictures of contact angles for different organic solvents.

Figure S7: Long-term test of filtration for the series organic solvents using CNT-EP-PC15 membrane.

Figure E: Surface SEM images of the membrane CNT-EP-PC15 after long-term OSN tests.

Question 9: *On Figure 4b please indicate clearly in the caption the thickness used for the calculation of the permeability. As a comment naming samples by code names makes the reading of the manuscript difficult and one should relate back to the synthesis conditions more often for clarity. Figure 4f could be updated with other relevant papers in this space and the benchmarking appears to be partial.*

Response: As suggested, we have specified the membrane thickness in the figure caption. We also updated the relevant papers in Fig. 4f (it is now Fig. 5f in the revised manuscript).

Question 10: *Overall the experimental details provided are insufficient and complete methods should be provided in supplementary materials.*

Response: We revised all the unclear descriptions in the method part.

Reviewer #2 (Remarks to the Author):

This manuscript reports a novel fabrication methodology, which enables membranes that exhibit impressive mechanical stability, promising organic solvent nano filtration performance, and molecular sieving capabilities. All the claims made are supported by convincing data and analysis. Further characterization of the PC-CNT interfaces would be a welcome (but not necessary) addition to this work as it obviously is the key for the high performance achieved. Considering that this is a highly competitive field of current and emerging significance, timely publication is essential. I recommend acceptance in its current form. I anticipate that this work will generate topical and broad interest as well as stimulate innovations and follow up studies.

Response: We sincerely appreciate the reviewer for the support of this work. We will continue studying the membrane interfaces and improving the membrane performance, as the reviewer suggested.

Reviewer #3 (Remarks to the Author):

The authors prepared a rigid Polycarbazole (PC) membrane on the polydopamin modified CNT support. Indeed, the high physical property including mechanical and chemical stability is an important issue in polymeric membrane because of the pressure-driven process. The membrane performance is quite impressive, although it is not the best. Recently, the authors reported the similar membrane in Nano Energy, and the membrane was applied in Li-S battery. The authors incorporated polydopamin in the previous CNT frame to PC membrane, and applied to solvent separation membrane. Overall, this manuscript has to be substantially modified to be considered in nature communication, as below.

Response: We appreciate the reviewer's precious time to provide this constructive review, which is of great help to improve the quality of this manuscript. We have made a major revision and provided additional results to address all the reviewers' questions. Detail explanations are provided in the below point-to-point responses.

Question 1: *Similar concept was recently published to Nano Energy by same authors, although it was employed in Li-S battery membrane. First of all, the recent paper should be cited in this manuscript, and should to be described the different between the system and component of the previous and present papers. "Electropolymerization growth of an ultrathin, compact, conductive and microporous (UCCM) polycarbazole membrane for high energy Li-S batteries" (Nano energy 2020)*

Response: It is our great pleasure to note that our work on Nano Energy has attracted the reviewer's attention. In that work, we used another type of conjugated microporous polymer (CMP)

membrane in Li-S battery to achieve the ionic separation between Lithium ions and polysulfides species. The basic idea is the same, that is, to utilize the uniform pore size of the CMP membranes achieved by the electropolymerization approach to separate challenging mixtures. However, there are two key differences. The first is that the ionic separation in Li-S battery is a concentration-driven but not a pressure-driven process. Without demonstrating the mechanic strength, the application of the CMP membranes will be significantly limited and thus the impact will be low. Second, the pore size for the separation of lithium ion and polysulfides species should be smaller than 1 nm, which is not suitable for organic solvent nanofiltration. As suggested, we quoted our *Nano Energy* work in Line 54-56.

Question 2: *Very importantly, the authors only showed the physical properties of the polydopamine modified CNT support. Because the physical properties of the composite membrane is more important, it is essential to show the physical properties of CNT-EP-PC composite membrane.*

Response: In the revised Fig. 2, Fig. 3, and Fig. S3, we provided various physical properties of the CNT-EP-PC15 composite membrane including SEM, TEM, AFM, XRD, pore size distribution measured by nitrogen physisorption, the mechanic properties by the standard tensile test and the advanced PFQNM method, and the surface wettability towards water and organic solvents.

Question 3: *Also, the authors should show the membrane performance in the pressure applied system.*

Response: We have made it clear in the revised manuscript that all the OSN tests and rejection of dyes were conducted under applied pressures. The only exception is the separation of mixed dyes in Fig. 5c, which was carried out in a transparent diffusion cell to visually demonstrate the separation. We added the applied pressures in the figure captions.

Question 4: *The PC film on the CNT frame exhibits narrow pore distributions. How about other PC-based membranes that have been reported previously? This has to be compared in the text and Figure 4f.*

Response: We noticed that in general the conjugated microporous polymers made of the EP process exhibited a narrow pore size distribution and a high surface area. However, all these results are reported in powder form, not in membrane form. The applications are also very different. Hence, it is not possible to make a meaningful comparison. However, we believe our approach can be extended to most PC-based polymer materials and thus will open a door for their broad applications in membrane separations.

Question 5: *Please explain why the CNT-polydopamine support shows the oriented dendritic structure. Are the structure and orientation of the CNT support influenced by the thickness and pore size distribution of the PC layer?*

Response: We believe the oriented dendritic structure of the PDA-CNT support is due to the sucking effect during the vacuum filtration. Yes, the pore size and porosity of the PDA-CNT support will influence the PC layer, mainly the mechanical strength. If the pore size and the porosity of the CNT support is too small, we won't be able to get a robust CNT-EP-PC composite. Our current understanding is that it is very critical to have a highly permeable support that allows the growth of the polymer from inside the porous structure to form a tight composite structure. Hence, we need home make the PDA-CNT support following the preparation procedure described in the method part, but not use the commercial ones. We have prepared hundreds of samples in this study, and it showed if the same procedure followed, the results were well reproducible.

Reference

1. Ravikovitch PI, Neimark AV. Characterization of micro-and mesoporosity in SBA-15 materials from adsorption data by the NLDFT method. *J Phys Chem B* **105**, 6817-6823 (2001).
2. Ravikovitch PI, Haller GL, Neimark AV. Density functional theory model for calculating pore size distributions: pore structure of nanoporous catalysts. *Adv colloid interface sci* **76**, 203-226 (1998).
3. Kupgan G, Liyana-Arachchi TP, Colina CM. NLDFT pore size distribution in amorphous microporous materials. *Langmuir* **33**, 11138-11145 (2017).
4. Pan YC, Li T, Lestari G, Lai ZP. Effective Separation of Propylene/Propane Binary Mixtures by ZIF-8 Membranes. *J Membr Sci* **390-391**, 93-98 (2012).

REVIEWERS' COMMENTS

Reviewer #1 (Remarks to the Author):

The work is well amended and I believe that it could be published

Reviewer #3 (Remarks to the Author):

The authors responded well, and I recommend to publish it as it is.